# Human stereoEEG recordings reveal network dynamics of decision-making in a rule-switching task

Marije ter Wal [1,2✉], Artem Platonov [3], Pasquale Cardellicchio [3], Veronica Pelliccia [4], Giorgio LoRusso [4], Ivana Sartori [4], Pietro Avanzini [5], Guy A. Orban [3] & Paul H. E. Tiesinga [1]

The processing steps that lead up to a decision, i.e., the transformation of sensory evidence into motor output, are not fully understood. Here, we combine stereoEEG recordings from the human cortex, with single-lead and time-resolved decoding, using a wide range of temporal frequencies, to characterize decision processing during a rule-switching task. Our data reveal the contribution of rostral inferior parietal lobule (IPL) regions, in particular PFt, and the parietal opercular regions in decision processing and demonstrate that the network representing the decision is common to both task rules. We reconstruct the sequence in which regions engage in decision processing on single trials, thereby providing a detailed picture of the network dynamics involved in decision-making. The reconstructed timeline suggests that the supramarginal gyrus in IPL links decision regions in prefrontal cortex with premotor regions, where the motor plan for the response is elaborated.

[1] Department of Neuroinformatics, Donders Institute, Radboud University, Heyendaalseweg 135, 6525 AJ Nijmegen, The Netherlands. [2] School of Psychology, University of Birmingham, Edgbaston B15 2TT, UK. [3] Department of Medicine and Surgery, University of Parma, Via Volturno 39E, 43125 Parma, Italy. [4] Claudio Munari Center for Epilepsy Surgery, Niguarda Hospital, Ospedale Ca'Granda Niguarda, Piazza dell'Ospedale Maggiore, 3, 20162 Milan, Italy. [5] Institute of Neuroscience, CNR, via Volturno 39E, 43125 Parma, Italy. ✉email: m.j.terwal@bham.ac.uk

During perceptual decision-making, sensory signals are mapped onto motor outputs, while integrating a multitude of factors, such as current motivation and goals, previous experience, risk, and potential reward. Many different processes (and brain areas) have been found to contribute to the preparation and formation of a decision[1,2], which have traditionally been divided into three functional groups: sensory encoding, decision representation, and motor output preparation[1].

Sensory evidence is built up by the relevant sensory cortices, as supported by changes in BOLD fMRI[3,4] and MEG/EEG[5,6] responses in humans, and selectivity of (subsets of) cells recorded in macaque sensory areas[7,8]. The preparation of the desired motor plans closely follows the accumulation of sensory evidence and can be read out in the relevant motor cortices, i.e., the frontal eye field and lateral intraparietal (LIP) area when eye movements are desired, or premotor and motor areas for hand movements[5,9–11]. LIP, which is situated in the intraparietal sulcus (IPS), has received most attention. In monkeys, LIP neurons increase their average firing rate with the accumulation of evidence[12,13] and sustain their firing until the response. The neural responses scale with, among others, the quality of the sensory information (e.g., easy versus difficult stimuli[3,14]) and task demand (e.g., speed versus accuracy[15]). LIP has therefore been proposed to encode the decision variable, i.e., the evidence for one or more possible decision outcomes within the task context, and the decision is made when LIP activity reaches a threshold[13]. However, recent findings point to a more complex representation of the decision variable: the dynamics of LIP neurons remain topic of discussion[16,17] and inactivation of response-related cell assemblies in LIP were shown to have variable effects on behavior[18].

Decision-related activity is also found in areas of prefrontal cortex (PFC) and posterior parietal cortex (PPC), beyond motor-related areas. These areas seem well suited to carry decision signals, due to their proximity to sensory-motor transformation regions and motor areas, and their shared involvement in task control[19]. However, the precise roles of PPC and PFC in the decision-making network remain topic of discussion[20]. fMRI work in human has demonstrated an abstract representation of the decision, independent of the motor system involved, in the dorsolateral prefrontal cortex (dlPFC), but not in PPC and other areas[3,21]. However, these findings rely on assumptions about the BOLD response and response contrasts between, for example, easy and difficult stimuli, which have since been questioned[14,16,22], as they do not take into account the non-monotonic relationship between neural recruitment and sensory evidence in monkey single-cell recordings[23]. The expected BOLD patterns are only partially observed in insula, inferior frontal gyrus, anterior cingulate cortex (ACC), and PPC[4,14,22], causing them to be under-reported as abstract decision regions. Using EEG, evidence accumulation signals independent of both sensory and motor systems[10,24] were found over central parietal cortex[10], potentially originating in posterior cingulate cortex[25]. Furthermore, single-cell recordings in monkeys suggest a role for both dlPFC and LIP in top-down regulation of decision-making with changing task rules[26]. However, it remains unclear whether these decision areas in PPC and PFC form a network that is shared between sensory features, task conditions, as well as motor outputs.

To address this question, a spatially and temporally precise insight into the decision-making process is required. Here, we analyze stereoEEG recordings from the cortex of drug-resistant epilepsy patients performing a rule-switching task. The rule-switching task allowed us to (1) identify decision signals (choice between left and right buttons), independent of the sensory content of the stimuli; and (2) determine whether decision signals are common to task rule and other task dimensions and if not, where they are modulated by those dimensions. Using linear classifiers, we identify a wide range of brain areas that were active in the stimulus-response interval, including ACC, dlPFC, and insula, which have previously been implicated for similar tasks. However, only a small subset of areas allowed us to successfully decode the left/right decision. The identified areas did not differ between task rules and stimulus features, suggesting they carry a common decision process. They include parietal operculum and supramarginal areas, which have so far gone unnoticed as decision-related areas, as well as known effector-related premotor and motor areas. To gain insight into the function of the newly identified decision areas, we analyze the onset of decoding on each trial for all decision areas. In line with previous work, we identify dlPFC and dorsal premotor cortex as carrying early decision information. Intriguingly, subregions of the supramarginal gyrus and parietal operculum also show early decoding of the decision, suggesting these too play a role in the elaboration of the decision.

## Results

**StereoEEG recordings during a rule-switching task.** Here we report data from six patients undergoing epilepsy monitoring in preparation of a focus-removal surgery at the Claudio Munari Center for Epilepsy Surgery of the Ospedale Niguarda-Ca' Granda in Milan, Italy (see Supplementary Tables 1 and 2 for demographic and diagnostic information). Patients had intracerebral electrodes stereotactically implanted[27] into frontal, temporal, and parietal regions of one hemisphere, allowing us to record local field potentials[28] from in total 663 gray matter contacts (Supplementary Table 3) that remained after removal of electrodes and trials with interictal epileptic activity. We refer to the "Methods" section for more information on recording and preprocessing procedures.

The patients performed a rule-switching task adapted from ref. [29] (see also ref. [30]). Patients had to judge the color (red or blue) or orientation (horizontal or vertical) of a bar-like stimulus (Fig. 1a), and subsequently press a button with either their left or right hand according to the decision table in Fig. 1a. Two of the four stimuli led to the same responses for both rules, i.e., were rule congruent, while the required responses to the other two stimuli were incongruent. The rule changed after a randomized period of 10–46 trials. The rule was cued by the color of an outline presented at the edge of the screen during a 300–600 ms period before stimulus onset. The cue and stimulus remained on the screen until the patient responded (Fig. 1b). The reaction times reported here represent the time between stimulus onset and button press. If the patient did not respond within 1500 ms of the stimulus onset, the trial timed out. Incorrect and timed-out trials were excluded from analyses.

Five of the six patients performed the task well (Fig. 1c), with decision accuracies between 86.7 and 98.5% (mean = 93.5%), while one patient had a performance level that only marginally differed from chance (54.8%, $p = 0.0358$, Chi-squared test, $\chi^2(1, N = 480) = 4.4083$), though this was mostly caused by trial time-outs (71% of excluded trials were time-outs). Decision accuracies did not significantly differ between left and right button presses (Fig. 1c; $p = 0.747$; two-tailed paired $t$-test; $t(5) = -0.3417$) and neither did accuracy between rules, congruent or incongruent stimuli, or the two colors and orientations of the stimuli (Supplementary Fig. 2), hence allowing us to compare left and right button presses across these task dimensions. Furthermore, reaction times did not differ between left and right button presses (Fig. 1d; left: $0.507 \pm 0.173$ s, $N = 1559$; right: $0.499 \pm 0.177$ s, $N = 1566$; $p = 0.216$; two-tailed $t$-test; $t(3123) = 1.238$), but they did differ for other dimensions of the task, notably rule (Fig. 1e; color: $0.540 \pm 0.139$ s, $N = 1578$; orientation: $0.511 \pm 0.137$ s, $N = 1547$; $p < 0.0001$; two-tailed $t$-test;

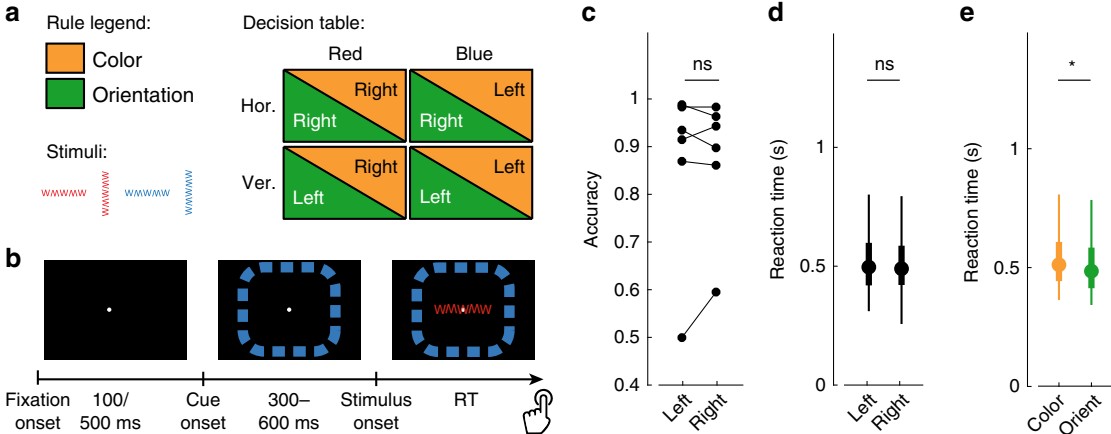

**Fig. 1 Task and behavioral performance. a** Stimuli and decision table showing the correct button press for each combination of rule and stimulus characteristics; **b** task sequence and screen layout; **c** task performance for left and right button presses. Each line connecting two dots represents one subject; **d** reaction time distributions for left (N = 1559) and right (N = 1566) button presses of all subjects combined (see Supplementary Fig. 2 for individual subjects); **e** reaction time distributions for the color (N = 1578) and orientation (N = 1547) rules. The boxplots in **d** and **e** show the 5, 25, 50 (filled circles), 75, and 95% boundaries. Gray horizontal bars represent the outcome of two-sided statistical tests of difference, using a paired *t*-test in **c** and Student's *t*-test in **d** and **e**. ns means not significant and * indicates significance at α = 0.01. Orange: color rule; green: orientation rule. **a** and **b** were adapted from ref. [30].

$t(3123) = 5.906$) and orientation (Supplementary Fig. 2; horizontal: $0.519 \pm 0.138$ s, N = 1573; vertical: $0.533 \pm 0.139$ s, N = 1552; $p = 0.0038$; two-tailed *t*-test; $t(3123) = -2.894$). The longer RTs for the color rule indicate that, as in the monkey[29], the color rule was more difficult. Reaction times showed an interaction between rule and color of the stimulus (Supplementary Fig. 2; $p = 0.0005$; two-way ANOVA, $F(1,3124) = 12.12$), probably caused by priming due to the colored rule cue. This interaction is expected to have little effect on the decision-related and response-locked analyses presented in the remainder of the paper. Subjects also responded significantly faster on correct than on incorrect trials (Supplementary Fig. 2; correct: $0.526 \pm 0.139$ s, N = 3125; incorrect: $0.597 \pm 0.183$ s, N = 170; $p < 0.0001$; two-tailed *t*-test; $t(3293) = -6.334$). In the remainder of the paper, only correct trials are included.

**Linear classifiers identify decision areas.** In order to identify brain regions involved in the rule dependent left/right decision-making process, we trained a linear discriminant analysis (LDA) classifier[31] to distinguish between trials, in which the patient pressed the left or the right button (see "Methods" section). A high classification performance represents either a large or a reliable distinction between the neural signatures of left and right trials, or both. Following a wavelet transform on the response-locked data, we used spectral power computed for 50 frequencies between 5 and 152 Hz as features for the classifier. This approach allowed us to (1) train the classifier on a single channel, preserving the high spatial precision; and (2) train the classifier individually for every point in time, preserving the high temporal resolution of the stereoEEG data. In addition, training the classifier on frequency-resolved data meant that we did not have to make assumptions about the frequency range of interest, and allowed for the possibility that for different brain regions information is contained in different frequency bands.

Across the six subjects, 95 out of the 663 gray matter leads showed significant response-locked decoding performance for the left/right decision (red circles in Fig. 2a; cluster statistic tested against 100 label-shuffled permutations with α = 0.05 and after false discovery rate (FDR) correction with $q = 0.10$, see "Methods" section). A large part of the identified leads was located in dorsal and medial premotor cortex and motor cortex

BA4, in line with the notion that decision information is represented by the relevant motor system[5,11], while leads in somatosensory areas BA3a/S1 likely carry feedback signals. In addition, the classifier identified three decision-carrying leads in the dlPFC, in agreement with earlier studies[3,21,26]. The classifier also decoded left/right decision in a small number of leads spread across the temporal lobe (TL), possibly due to the visual nature of the stimuli. Intriguingly, however, a third of the leads showing successful left/right decoding were located bilaterally in the supramarginal gyrus and parietal operculum, areas that have so far gone largely undetected as decision areas. We further localized the leads into cytoarchitectonic areas PFt, PFcm, PF, and PFop of the supramarginal gyrus, and parietal operculum areas OP1–OP4 (see Supplementary Fig. 1 and the "Methods" section for details on the localization, and Supplementary Fig. 3 for the number of leads per area and per patient). Further results concentrate on the 12 areas sampled by at least three significant leads (86 leads in total).

To ensure that the left/right classification results were functionally meaningful, we tested whether the same leads were activated by the task, using two different approaches. Firstly, we trained a new set of time-resolved classifiers to distinguish active trials from baseline activity (see "Methods" section). We did this for all 663 leads, of which 211 showed significant activity in the response period (shown as blue dots in Fig. 2a). Note that there is a large overlap between the left/right classifier and the activity/baseline classifier, with 90.5% of left/right leads showing significant decoding of activation over baseline in the response period ($p < 0.0001$; Fisher's exact test). The activity/baseline classifier also revealed several brain areas that showed task-related activity, but did not carry significant left/right information, e.g., insula, fusiform and parahippocampal gyri, and anterior cingulate.

Secondly, we compared the left/right decoding performance with the power spectra of the identified leads. When contrasting trials with contralateral button presses, relative to the recorded hemisphere, against ipsilateral trials, we found substantial power differences in both the (pre)motor, sensory, as well as the PF and OP areas (Fig. 2b, Supplementary Figs. 6 and 7), supporting the result from the classifier that PF and OP areas carry left/right information. Note that for the power spectra, as with the

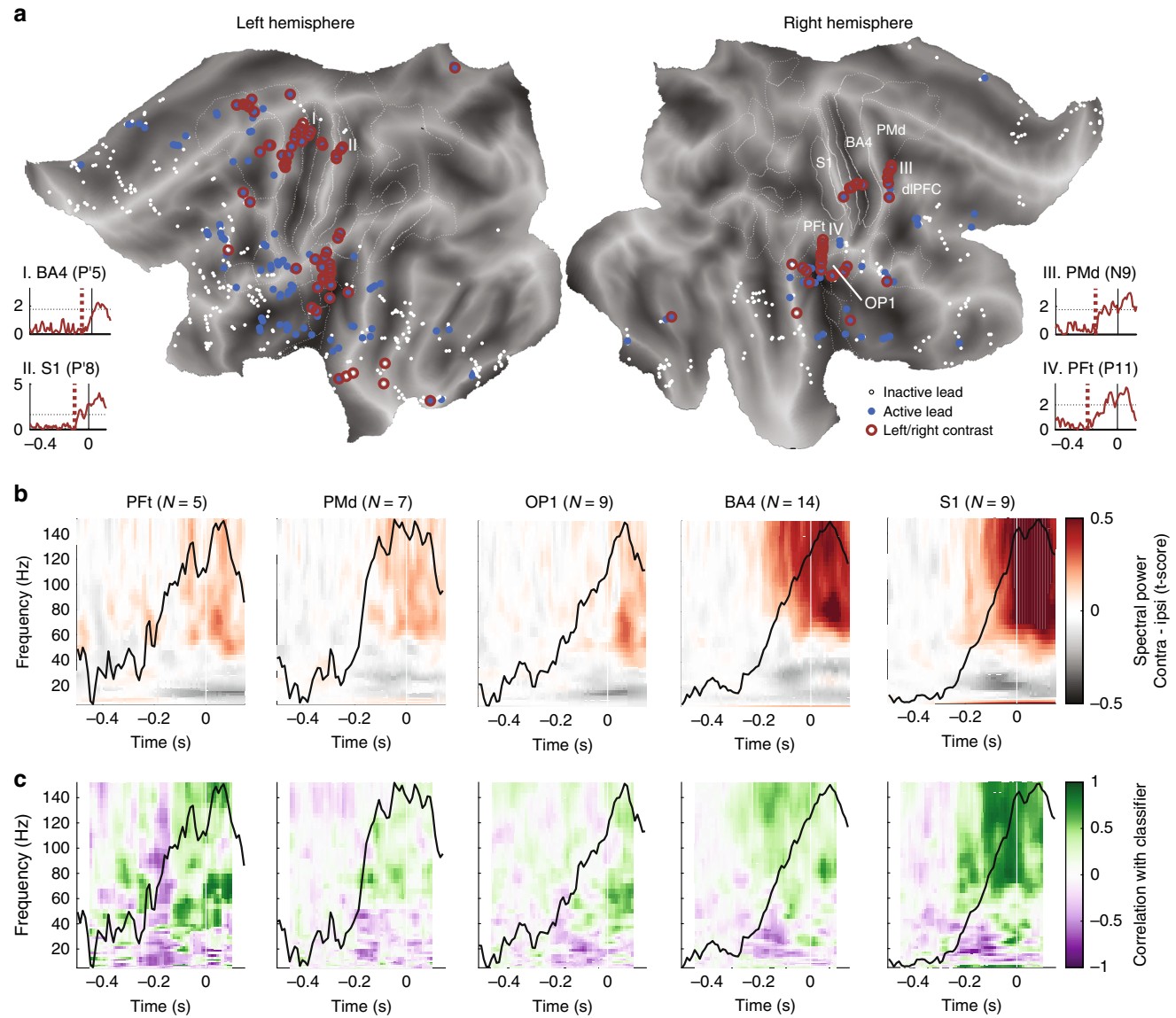

**Fig. 2 LDA classifier identified leads that carry decision information. a** Flatmaps showing, for all recorded leads across six patients whether a significant response-locked change from baseline was detected (blue dots) and/or the left/right decision was successfully decoded (red circles), or whether their activity did not differ from baseline in the interval starting 0.5 s before and ending 0.15 s after the response (white dots). Brain areas of interest are indicated by white outlines. The insets show the *t*-scored classification performance of four example left/right decoding leads (indicated by I–IV on the flatmaps). Dotted lines indicate the onset of temporal cluster of the significant classification. For full labeling of areas see Supplementary Fig. 1; **b** average spectral power per area (number of leads given in brackets), including only those leads that showed significant decoding of the left/right decision. To combine data recorded in both hemispheres, data were contrasted as contralateral (i.e., left button presses when recording in the right hemisphere) versus ipsilateral decisions (i.e., right button presses when recording in the right hemisphere); **c** average correlation per area between spectral power and classification performance across all left/right decoding leads. Correlations were computed using a 100 ms wide sliding window. Green indicates that power increase related to high classification performance and purple indicates that power decrease co-occurred with high classification performance. In **b** and **c**, black lines give the average response-locked classifier performance across leads in the area, normalized to the peak performance (*y*-scale between 0 and 1). Non-scaled performances per lead are given in Supplementary Fig. 4, stimulus-locked and response-locked classifier performance are compared in Supplementary Fig. 5, and spectral power and power-classifier correlations for all other areas of interest can be found in Supplementary Figs. 6–9.

classifier, we used a left/right contrast to identify regions that represent the decision. However, to pool these results across subjects, we converted this to a contra-ipsilateral hemisphere contrast, as two subjects were implanted in the right hemisphere and four subjects in the left hemisphere. Spectral power for contra- and ipsilateral trials are given separately in Supplementary Fig. 7.

We next asked what part of the spectral content of the signals informed the left/right classifier at each time point; in other words, what is the spectral signature of the left/right decision in

each brain area? To assess this, we computed the correlation between the classifier performance and the spectral power contrast for each lead, and frequency across time in 100 ms sliding windows (Fig. 2c). Note that the LDA classifier used here is a multivariate approach and can use any linear combination of frequencies to classify the trials. Therefore, a lack of correlation between classifier and power at a specific frequency does not indicate a lack of information at that frequency in general, but only a lack of univariate correlation, while a high correlation can be taken as indication that a given frequency provides left/right

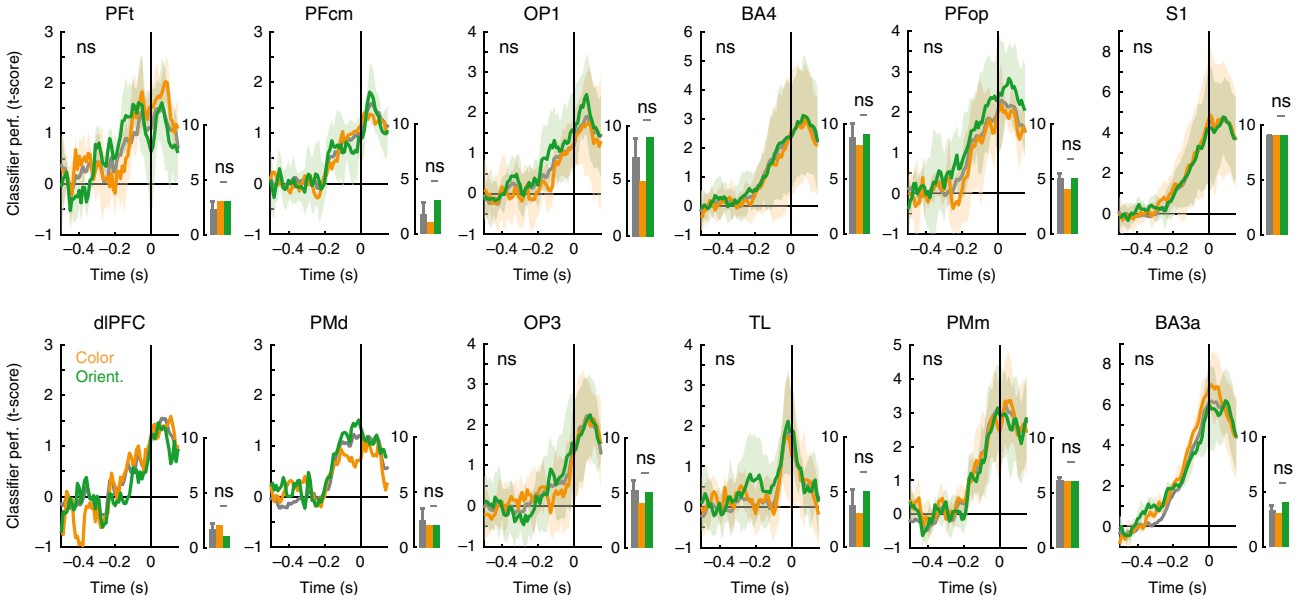

**Fig. 3 Decisions for color and orientation rules invoke similar responses in the same brain regions.** The left panels show the average classifier performance traces (± standard deviation, numbers of data points included are given in the right panels) for classifiers trained on only color rule trials (orange), only orientation rule trials (green), and the average of ten subsampled trial sets that included both color and orientation rules, and match the trial count of one rule (gray). All brain regions that showed significant left/right decoding based on all trials are included here. For all regions with three or more decoding leads for both the color and orientation only classifiers, a cluster-based permutation test was performed to compare decoding performance of color and orientation over time. None of the clusters survived FDR correction. The right panels show the number of leads that show significant decoding for each of the classifiers. For the subsampled classifier bars show the mean, and error bars indicate the standard deviation across ten repetitions (raw data can be found in Supplementary Table 5). Horizontal bars indicate a one-sided Wilcoxon rank test of the difference between the number of leads for color and orientation rule compared to pairs of randomly subsampled classifiers ($\alpha = 0.05$; FDR corrected); ns is not significant. The total number of leads per brain area can be found in Supplementary Fig. 3b.

information. Across all brain areas, we found a strong positive correlation between gamma power (>50 Hz for premotor, >30 Hz for other areas) and classifier performance around the response time and negative correlations with alpha/beta power (10–30 Hz), indicating that response information was associated with high gamma power and reduced alpha/beta power. Intriguingly, the PF and OP areas showed an additional negative correlation between classifier and gamma frequencies at earlier time points (Fig. 2c, Supplementary Figs. 8 and 9). This suggests that the PF and OP areas undergo a decision-predictive initial reduction in gamma power in the contralateral hemisphere, and/or an increase in power on the ipsilateral hemisphere, followed by an increase in gamma power in the contralateral hemisphere (Supplementary Fig. 7). The non-monotonic response points to complex, biphasic dynamics in the PF and OP regions.

**Decision network is agnostic to task dimensions.** Our classification and spectral analyses suggest that an extensive network of brain regions is involved in the preparation and/or execution, and evaluation of the decision. Our task design allowed us to test whether this network was specific to stimulus or task characteristics, namely stimulus color and orientation, task rule and rule congruency. Rule information has been shown to be represented at the stage of dlPFC[26], one of the regions showing left/right decoding here. Does the left/right decoding network differ for different task rules? To test this, we retrained the left/right classifier on color rule and orientation rule trials separately. This halved the number of trials available to the classifier and to compensate for this, we also retrained the classifier on ten randomly subsampled trial sets containing trials from both rules. The total number of leads showing significant decoding did not differ between color and orientation rules ($p = 0.289$, Fisher's exact test)

and neither did the number of leads in any of the recorded brain regions (Wilcoxon rank-sum tests against 45 pairs of subsampled trial sets; right panels of Fig. 3 and Supplementary Fig. 10a). Furthermore, time traces of the color rule and orientation rule classifiers showed no differences for any of the brain regions with three or more leads (left panels of Fig. 3; cluster-based *t*-test with 500 permutations and FDR correction, see "Methods" section). These findings indicate that the two rules engaged identical brain regions, with similar time courses, for the elaboration of the decision. Similarly, we only found differences in number of leads showing significant decoding for congruent and incongruent trials for OP1 and somatosensory areas (Supplementary Fig. 10b), while red and blue trials, and horizontal and vertical trials only differed in S1 (Supplementary Fig. 11).

**Analysis of classifier onset times.** What is the functional role of the newly identified parietal PF and OP areas in the decision-making process? To shed light on this, we analyzed the precise timing of the responses in the information-carrying areas. As response times can be biased when using a classifier performance metric across trials, for example, in areas sensitive to trial-to-trial variations in difficulty level, we instead identified response timings in single trials. To this end, we obtained a decoding performance trace for each trial individually, by retraining the LDA classifiers using a leave-one-out cross-validation approach for all the previously identified left/right decoding leads (see "Methods" section for details). The leave-one-out classifier provided us with a decision value (D-value, see "Methods" section), a proxy for the certainty with that the time point in the trial could be decoded by a classifier trained on the same time point from all other trials. For each trial, we then identified the time point at which the classifier performance started to increase toward the highest

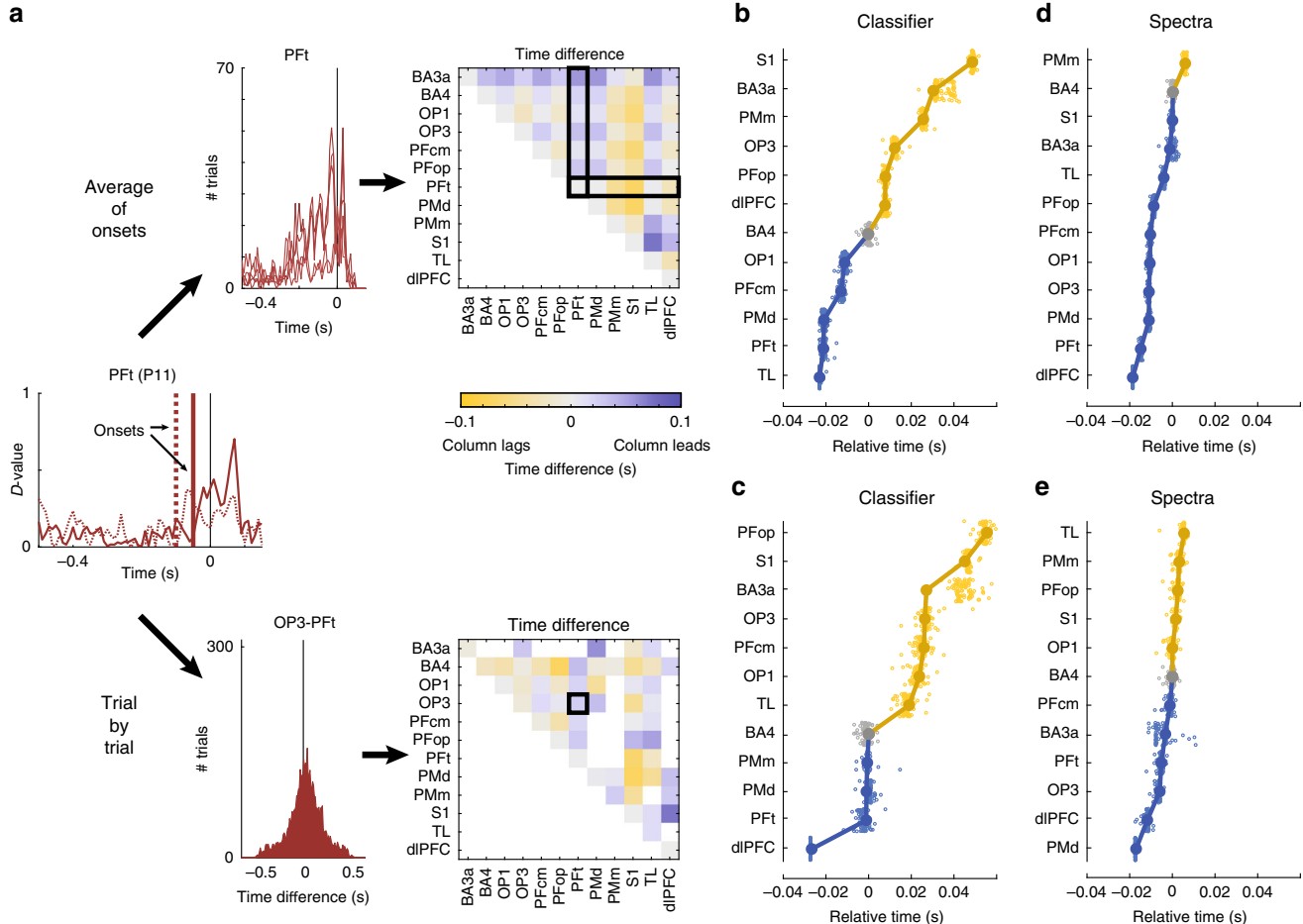

**Fig. 4 Single-trial analysis of activity onset for classifier and spectral power traces. a** Schematic explanation of single-trial onset analysis. For every single trial the onset of the highest peak of classifier performance ($D$-values) or single-trial power trace was determined. Here, two example classifier $D$-value traces are shown for lead P11 located in area PFt, with onsets indicated by vertical lines. Subsequently, either the average onsets are computed across all leads in each area of interest, after which the differences of the average onsets is determined ('average of onsets'), or the differences in onset of pairs of leads are compared for each trial, after which the differences are averaged within the areas of interest ('trial by trial'). Both methods thus resulted in a difference matrix between all areas of interest, which was then ordered on a linear time axis using a multidimensional scaling approach. For power traces (not shown in this example), onset analyses were computed for each frequency independently and time difference distributions were combined across frequencies; **b–e** temporal ordering results of the average of onsets approach (**b**, **d**) and the trial-by-trial approach (**c**, **e**) for single-trial classifiers (**b**, **c**) and spectra (**d**, **e**). Connected filled circles indicate the onset of each area relative to the onset of area BA4 (gray), with blue areas starting before BA4 and yellow areas appearing after BA4. Small dots represent bootstrapping results, with each bootstrap leaving out the data from one recording lead ($N = 86$). The classifier onset distributions of all leads can be found in Supplementary Fig. 12 and time difference matrices for all four methods are given in Supplementary Fig. 13.

performance peak (Fig. 4a left). This provided us with a distribution of trial-specific onset times for each lead.

We then obtained onset time differences for all pairs of identified brain areas, using two different strategies. In the first approach (Fig. 4a), we combined the onset distributions of all leads in one brain area and computed the average onset for this area. We then computed the time difference matrix by comparing onsets of all pairs of brain areas. Averaging within a brain area provides a more stable estimate of the onset and has the advantage that the difference matrix is complete, but note that a difference is computed even if a pair of areas is not represented in the recordings from a single patient. To account for this, we used a second approach (Fig. 4a), where we computed the time difference between each trial individually for every pair of leads recorded simultaneously. We then collapsed the onset time differences within brain areas and averaged across subjects. Note that the resulting time difference matrix is incomplete when pairs of brain areas were not recorded in any of the subjects.

To aid interpretation of the time difference matrices, we projected them back onto a timeline using a simplified multidimensional scaling (MDS) approach. This approach can only produce a relative timeline and not the "absolute" time before the response, as the absolute time-to-response is subtracted out in the pairwise comparisons. We therefore translated the timelines to the onset of motor cortex (BA4), as a proxy for response onset. We validated each timeline with a bootstrapping procedure, leaving out the results from one lead from each bootstrap (leading to 86 bootstraps per method). The resulting timelines and their bootstrapping results for the average onset and trial-by-trial approaches are shown in Fig. 4b, c, respectively. Both approaches identified early decoding onsets for area PFt and dorsal premotor area, together with dorsolateral prefrontal dlPFC for the trial-by-trial method, with all three areas having a response onset well before motor cortex. Though the exact timing differed slightly between methods, onsets in PMm, PFcm, OP1, and OP3 roughly coincided with BA4 response onset in

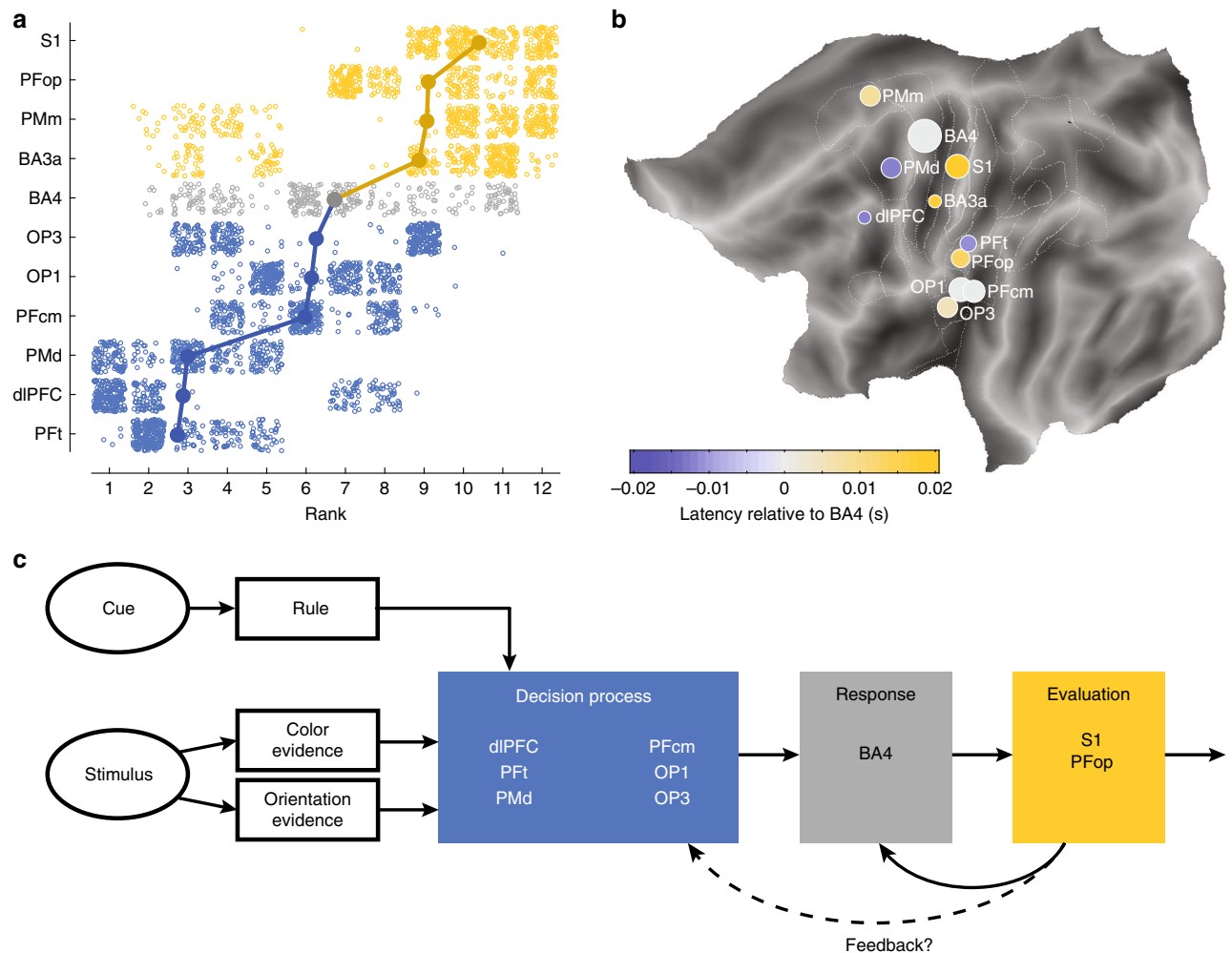

**Fig. 5 Integration of the results of the four onset-tracking methods. a** Rank distributions (i.e., ignoring the numerical value of onset) combined for all four methods from Fig. 4b–e, with each dot representing one bootstrap ($N = 86$) from one method ($N = 4$). All areas shown here achieved significant conjunction. Areas are ordered by their average rank; **b** average onset time of activation (color coded), relative to onset of area BA4, for all areas with significant consistency between methods. Each dot is located at the center of mass of all recording leads that contributed and the size of the circle represents the number of recording leads; **c** schematic of information flow suggested by the results in Figs. 3, and **a** and **b**: stimulus information and rule information all converge onto a shared decision process, consisting of an early (dlPFC, PFt, and PMd) and a late phase (PFcm, OP1, and OP3). See "Discussion" section for a detailed description.

both. Sensory areas BA3a and S1 were last to classify the response, together with area PFop.

As the classifier performance is based on a wavelet decomposition using frequencies between 5 and 152 Hz, the time differences between areas can be confounded by differences in frequency content, with lower frequencies leading to broader responses and hence earlier onsets. To exclude this possibility, we performed the analysis described for the classifier performance on the contra-/ipsilateral power contrast traces for each frequency (see Fig. 2b for the average contrasts). To capture the potentially biphasic power response that we described earlier for PF and OP leads, we detected both increases and decreases in power contrast. By only averaging across frequencies after relative times have been computed, any influence of wavelet width is removed. Though the single-frequency single-trial analysis is much more variable and lacks the multivariate information used by the classifier, we still obtained timelines qualitatively similar to those obtained from the single-trial classifiers (Fig. 4d, e).

**Consensus confirms early role for dlPFC, PMd, and PFt.** To test the consistency and reliability of the timelines, we integrated the

four methods into a single timeline. Due to differences in scale of the time axes, the timelines will have different weights when computing a standard average. We therefore opted for a non-parametric approach, by identifying the rank order for each bootstrap for every method and collapsing the rank distributions across methods (Fig. 5a). We then computed a rank clustering score for each area (see "Methods" section) and compared this to 500 randomly generated datasets. All identified brain areas showed significant clustering ($p < 0.002$), except for the collection of TL leads ($p > 0.998$), which is therefore excluded from Fig. 5.

Both the integrated rank (Fig. 5a) and the average onset across methods (Fig. 5b) paint a picture of early involvement in decision-making by areas PFt, dlPFC, and PMd, with these areas responding, on average, ~12 ms before area BA4. The response onset of BA4 coincided with the onset of PFcm, OP1, and OP3. Areas PFop and PMm showed late responses, together with sensory areas BA3a/S1, with the onset of S1 responses following ~25 ms after the onset of BA4. The late responding BA3a/S1 areas also stand out due to their high decoding performance (Supplementary Fig. 4). This suggests that for decision regions, there is little or no evidence that earlier onset is coupled to the highest decoding performance (Supplementary Fig. 14).

Patient 6 did not perform the task well, yet this had negligible impact on the results, as demonstrated by extremely high correlations for rank (Spearman; rho = 0.991; $p < 0.0001$) and timing (Pearson; rho = 0.994; $p < 0.0001$) between the full dataset and a reduced dataset without their data (Supplementary Fig. 15a).

The majority (567/663) of tested leads were located outside the epileptogenic zone (EZ), yet 16 of the 95 left/right classifier leads were located in the EZ. Inclusion of the latter leads in the results is unlikely to influence the left/right classification results. Indeed, the fraction of leads with significant classification performance did not differ inside and outside of the EZ (Fisher's exact test; $p = 0.528$ for the left/right classifier; $p = 0.556$ for the baseline classifier). When computed without the EZ channels (Supplementary Fig. 15b), the integrated timeline produced similar results to the full dataset, leading to high correlations between the datasets for the rank conjunction (Spearman correlation; rho = 0.861; $p = 0.0003$) and onset times (Pearson; rho = 0.839; $p = 0.0007$).

## Discussion

By performing single-lead decoding on human stereoEEG data, we have identified brain areas carrying decision information in a rule-switching task. We reported significant classification performance in areas in the parietal operculum and the supramarginal gyrus, which have so far remained undetected in studies using BOLD fMRI and EEG/MEG in humans, or using invasive recordings in monkeys. In addition, we confirmed the involvement of previously identified brain areas in early decision-making processes, such as the dlPFC and premotor areas, and were able to decode the button press from somatosensory areas, such as S1. Leads showing successful decoding did not differ between the two task rules, indicating that the identified regions carry a task-independent representation of the decision (Fig. 5c). To investigate the function of the newly found decision areas, we assessed when decision information appeared in each area, by determining the onset of successful decoding with the LDA classifier and power differences between left and right trials. Both measures indicated early onset of decision information in OP and PF areas, with area PFt responding first, together with dlPFC and PMd, while PFcm and OP areas follow around the onset of BA4 (Fig. 5c).

The early onset of decision information in dlPFC aligns with its proposed abstract representation of the decision, i.e., independently of the motor plan required for output[3,21]. However, more recent studies suggest that abstract information is also present in human parietal areas[10,24], and that macaque PFC and LIP both carry early decision information, as well as top-down task information[26]. The timing of posterior parietal areas and dlPFC presented here, using a similar task, agrees with the notion of an early decision network shared between PPC and PFC; including, however, a previously unnoticed PPC region, the supramarginal gyrus (Fig. 5c). The comparison between classification performance and spectral power contrast suggests that the dlPFC and supramarginal areas (Fig. 2c) do not show a gradual, monotonic increase leading up to the decision, as was previously reported for a human posterior parietal region[14,22,23]. Instead, these areas show a non-monotonic response with an initial negative correlation between classification performance and power at higher frequencies, followed by a late positive correlation at these frequencies. This shared electrophysiological signature can point to a close functional link between supramarginal gyrus, in particular PFt, and dlPFC. Indeed, anatomical connections between the anterior PF region of the intraparietal lobule, and both PFC and dorsal premotor areas are well documented in macaque[32] and human[33].

Our results suggest that one of the supramarginal areas, PFt, is in a position to link the decision region in the PFC with premotor regions. PFt may therefore correspond to the abstract decision step (independent of required motor output) identified by O'Connell et al.[10] and proposed to be located in rostral parietal cortex (but see ref. [25]). Although our dataset did not include electrodes placed in the more posterior IPS, and hence the timing and signature of IPS responses cannot be compared directly to our stereoEEG results, previous work suggests that IPS operates in decision task on a similar time frame as dlPFC[26]. Hence a plausible sequence of events is that IPS encodes the decision variable based on sensory inputs[18,26], in order to track and coordinate evidence accumulation. The decision information from IPS is sent, online and parallel to sensory information, to dlPFC where it is transformed into a task-relevant representation[19], and to PFt to be transformed into a motor-compatible representation. The motor-compatible representation can then be read out and converted to an all-or-none response in the pre-motor areas[34], at which level speed-accuracy trade-offs may be incorporated[35]. The transformation from a task-relevant to motor-compatible representation of the decision variable might be particularly important in the rule-switching task we used here, in which stimulus features did not directly map onto button presses due to rule-dependence of the desired output[36]. Our findings hence put new emphasis on the need for a comprehensive model of the steps and transformation needed to reach a decision[20].

Other supramarginal regions, in particular PFop, followed later in the sequence, after BA4. They are suited for a role in post decision processing[37], and in particular compute a perceptual measure of confidence by comparing accumulated evidence for the two choices at the moment of decision[38]. This signal can be further elaborated into confidence when transmitted to the PFC (ref. [39]; Fig. 5c).

Like the supramarginal gyrus, the parietal operculum has also remained undetected as decision-related area. All four OP subdivisions respond to tactile stimulation[40], though OP1 does so most consistently. OP1 has been reported to show both phasic and tonic responses to contralateral median nerve stimulation[41], and tonic responses to ipsilateral nerve stimulation[42], suggesting the left/right decoding from OP1 reflects phasic tactile inputs. OP includes the secondary somatosensory area (SII)[40], which has bidirectional connections with S1 (ref. [43]). SII has been reported to play an important role in tactile decision-making[44], possibly carrying the decision variable, but to our knowledge OP activation in decision-making has not been reported for sensory modalities other than tactile. The present results suggest that the involvement of OP1 reflects tactile feedback (Fig. 5c), just as SI, but combined with proprioceptive feedback[45] and possibly predictive signals about the feedback. Beyond this possibility, the role of OP areas in the rule-switching task remains to be established.

Why have the supramarginal and parietal operculum areas remained undetected as decision areas in previous studies? We hypothesize this partly reflects technical limitations. PF and OP are relatively deep areas, and their convex shape and the proximity of more superficial concave gyri reduce their detectability in EEG recordings[28]. Depth is also likely to affect MEG detection, in addition to a suboptimal orientation of the gray matter in OP and PF relative to the skull[46], though OP regions have been targeted using MEG[47]. On the other hand, BOLD fMRI studies have successfully recorded OP[48] and PF activation (ref. [49]; and many others). Yet, in decision paradigms, inferior parietal lobule activation has, to our knowledge, only previously been mentioned for a perceptual recognition task[2]. One possible explanation is that the BOLD response is not suitable to pick up the fast dynamics of these rostral parietal regions (i.e., switch in correlation between

power and classifier performance in Fig. 2). Furthermore, it is unlikely that representations in parietal operculum and supramarginal gyrus scale with stimulus features, a contrast often used to select decision areas in BOLD fMRI studies[3,4,14,22].

In this study, we obtained classification results with single-lead resolution by using frequencies as features. So far, classification of intracranial data has mostly been performed with leads as features[50,51], allowing for high temporal, but low spatial resolution. Leaving out one dimension (time, frequency, or electrode location) has been used to estimate the contribution of each dimension[52,53], but this requires specific assumptions about the timing and location of the effect under study.

In addition to using a single-lead classifier, we trained the classifier for each time point individually. As we used a leave-one-out cross-validation approach, this allowed us to assess the time course of classification on individual trials for each lead. Time-resolved classification has already provided insights in other fields, e.g., the timing of retrieval of different aspects of memories[54] and visual processing[55], and was identified as an important step forward in informing models of decision-making[56]. Here, we have shown that single-trial decoding can give unique insight into the decision-making process, emphasizing the role of the supramarginal gyrus. Further work, recording from more patients and covering more posterior parts of the cortex, is needed to clarify the precise role of the diverse parietal and prefrontal players in the decision process.

## Methods

**Patients and implantation procedure**. Data were collected from six drug-resistant epileptic patients (one female, aged 19–44 years). Detailed demographic and diagnostic information about the patients can be found in the Supplementary Tables 1 and 2. Patients presented either no anatomical alterations ($N = 5$) or alterations located outside the regions of interest ($N = 1$). Data from patient P01 and P02, who participated in the pilot experiment, were previously reported in ref. [30]. All patients were admitted to the Claudio Munari Center for Epilepsy Surgery of the Ospedale Niguarda-Ca' Granda in Milan, Italy, for identification of the epileptic focus and subsequent surgical removal. Patients were stereotactically implanted with 12–17 cylindrical probes (DIXI Medical, Besançon, France), which contained 8–18 contacts (or leads) of 2 mm length spaced 1.5 mm apart. Implantation sites were selected by the medical team on clinical grounds, according to ictal semiology, scalp-EEG, and neuroimaging studies, and with no reference to the experimental protocol presented here. Signals were sampled at 1000 Hz using a Neurofax EEG-1100 system (Nihon Kohden), along with a trigger channel and two electro-oculography (EOG) electrodes. The stereoEEG electrodes were referenced against the average signal of two adjacent leads, both located in white matter. The reference leads were selected for each patient individually, based on the requirement that they did not show any response to standard clinical stimulation, including somatosensory (median, tibial, and trigeminal nerves), visual (flash), and acoustical (click) stimulations, nor did electrical stimulation evoke any sensory and/or motor behavior, (see also ref. [41]). The Ethics Committee of Ospedale Niguarda-Ca' Granda (ID 7-012013-25.01.2013) approved the study. Patients were fully informed of the stereoEEG implantation and recording procedures, and gave their written informed consent to participate in the study, according to the Declaration of Helsinki (BMJ 1991; 302:1194). Recordings took place in two sessions during the seizure-free postimplantation period.

**Task**. The patients performed a rule-switching task based on the task presented in ref. [29] (Fig. 1a, b). Each trial started with the presentation of a cue, following an intertrial interval in which only a fixation cross was shown. The cue consisted of a colored band around the edge of the screen. A red outline indicated that the color rule was active, whereas a blue outline cued the orientation rule. The duration of the cue period was randomized and chosen from 18 intervals uniformly spanning the 300–600 ms range. After this interval, the stimulus appeared in the center of the cue outline, while the cue remained on the screen. Stimuli consisted of a series of W's and M's, forming a bar-like structure. Stimuli were always either red or blue, and were either horizontal or vertical (Fig. 1a). Red-horizontal and blue-vertical stimuli always required right, respectively left, button presses, i.e., these stimuli were rule congruent, while red-vertical and blue-horizontal stimuli were rule incongruent. The rule switched after a randomized period of between 10 and 46 trials. Subjects were instructed to respond as accurately as possible soon after stimulus onset. If the subject did not respond within 1500 ms after stimulus onset the trial timed out. After the subject's response the fixation cross reappeared, indicating the start of the intertrial interval. The patients completed two blocks of trials. The first two patients did two blocks of 400 trials each, with intertrial

intervals of 100 ms, while the other four patients completed blocks of 240 trials each, with longer (500 ms) intertrial intervals, to allow for a better estimate of low frequency content. As these differences did not affect the analyses presented here, results from all six patients were combined.

**Localization of electrodes**. The location of each electrode was determined as described in ref. [41], based on postimplantation CT scans (O-arm scanner, Medtronic) co-registered to preimplantation T1-weighted MRIs (voxel size $0.5 \times 0.5 \times 2$ mm). Subsequently, multimodal views were constructed using the 3D Slicer software package[57], and the exact position in the brain of all leads implanted in a single patient was determined using multiplanar reconstructions and Freesurfer[58] computed surfaces.

Leads located in white matter were excluded from further analysis. In total, between 92 and 141 gray matter contacts were recorded in each patient (663 leads total, see Supplementary Table 3). Gray matter lead locations were imported into a common template, using the warping of the individual cortical anatomy to the fs-LR template[59]. Leads were assigned to the cytoarchitectonic brain regions imported from the Anatomy Toolbox (https://www.fz-juelich.de/inm/inm-1/DE/Forschung/_docs/SPMAnatomyToolbox/SPMAnatomyToolbox_node.html)[60] onto the template (see Supplementary Fig. 1 for full labeling of regions), following the work of Katrin Amunts, Karl Zilles and colleagues[61–64]. We combined the results from leads in areas BA1, BA2, and BA3b into area S1. Areas with fewer than three leads were not included in the analyses for Figs. 2b, c, 4 and 5. Of the gray matter leads, between 3 and 30 leads per patient (96 leads total, see Supplementary Table 3) were identified as located in the epileptogenic zone (EZ). Control analyses for the impact of the EZ electrodes are reported in the "Results" section.

**Data preprocessing and artifact rejection**. To facilitate artifact rejection, we transformed the data to the time–frequency domain by convolving with complex Morlet wavelets at 50 ms time intervals, using the cwt routine from the Wavelet Toolbox for MATLAB. The wavelets were scaled to approximate frequencies between 50 and 150 Hz at increments of 10 Hz (ref. [41]). The quality of the data was visually inspected using plots of gamma power time course in all trials collected for a given condition, to detect the possible presence of ictal epileptic discharges (IEDs). All trials/channels in which any IED or other transient electrical artifacts appeared were removed. The numbers of removed items are listed in Supplementary Table 4.

The raw data from artifact-free trials were band-pass filtered between 1.5 and 300 Hz using a sixth-order Butterworth filter. Line noise (50 Hz) and its harmonics were removed using Notch filters with a 3 dB power reduction at ±0.02 Hz.

Trials for which the subjects did not respond in time, or for which they answered incorrectly, were not included in the analyses presented in this paper. For five of the six subjects, the number of correct trials significantly exceeded chance level (86.7–98.5% correct, see Fig. 1c and Results section), but the sixth subject only answered 54.8% of trials correctly, which only marginally differed from chance ($p = 0.0358$, Chi-squared test, $\chi^2(1, N = 480) = 4$). This subject answered 12.9% of the trials incorrectly and the remaining 32.3% timed-out. The main results include all six subjects; however, control analyses limited to the five above-chance performing subjects produced the same outcome (Supplementary Fig. 15).

In order to prevent classification based on eye movement-induced signals, particularly in frontal leads, an independent component analysis (ICA) was performed on the data of each patient, using the runICA.m implementation from EEGlab[65]. Components that correlated significantly with the electro-oculography (EOG) channels (5% of components) were removed and the reconstituted data were subsequently transformed back to channel space.

**Wavelet analysis**. All remaining trials were wavelet transformed by using a complex Morlet mother wavelet of four cycles wide at 10 ms time intervals, using the continuous wavelet transform function cwt from the Wavelet Toolbox for MATLAB. Wavelets were scaled to cover frequencies between 5 and 152 Hz at 50 semilogarithmic intervals. The absolute value provided an estimate of spectral power. We then cut the trials into epochs starting 0.5 s before the button press to 0.15 s after button press. To report the spectral signature of the left/right decision (Fig. 2b), we contrasted the trials where the button contralateral to the recorded hemisphere was pressed to trials with an ipsilateral button press, by computing the $t$-statistic for every time point and frequency. For interpretability, we also $z$-scored the power spectra against a 500 ms pre-cue baseline for contralateral and ipsilateral trials separately (Supplementary Fig. 7).

**Left/right and baseline classifiers**. To identify leads that carried information about the left/right decision, we trained LDA classifiers[66] on the wavelet transformed and response-aligned data. We used the power per wavelet scale as features, allowing us to train a classifier for each time point and for each lead individually, preserving the high spatial and temporal precision of the data. We did not include phase in the feature space[67], as exploratory analyses suggested including phase did not improve decoding. To minimize overfitting, we used shrinkage regularization and a $k$-fold approach to cross-validate the classifier's performance, with five folds. The selection of the folds was repeated ten times. For every lead and time point, decision values for the left and right classes were combined to a $t$-statistic

representing classification performance across all trials. We used the LDA classifier implementation from the MVPA-Light toolbox used in ref. [31] and available on (https://github.com/treder/MVPA-Light).

In addition to identifying differential activation between left and right trials, we also aimed to identify general response-related activation, including activation that is shared by both left and right button presses. To this end, we created a 'baseline' dataset by shuffling the time points within each trial and for each frequency. We trained classifiers on the shuffled versus the intact trials. The classifier parameters were otherwise identical to the left/right classifier.

**Identification of temporal classification clusters.** To obtain empirical reference distributions for comparison with the classifier performance, we randomly shuffled the class labels (either for left versus right contrast or for intact versus time shuffled contrast) and retrained the classifier on the shuffled labels. Label shuffling and retraining was performed for each lead individually and repeated 100 times, thus generating a lead-specific estimate of chance level that takes into account any structure in the dataset, other than the class labels. To minimize influence from other task dimensions, specifically, task dimensions rule, color, and orientation, we required that an approximate balance for these task labels was retained on each permutation. For example, stimulus color was balanced for the intact labels, so 50% blue and 50% red, within each rule. When the class labels were shuffled, we required that each shuffled rule class contained at least 30% blue or 30% red trials. We treated the other task dimensions in the same way.

Each classifier was trained for individual time points spaced 10 ms apart. Yet we expect classification of the left/right decision to persist for an extended period around the response, so time points should not be considered in isolation. We therefore identified temporally contiguous clusters[68] of significant classification performance by thresholding the classification $t$-statistic trace for each lead at $\alpha = 0.05$ and the number of degrees of freedom defined as df $= N_{correct}^s - 2$, with $N_{correct}^s$ the number of correct trials for subject $s$ (Supplementary Table 4). For each period of threshold crossing, we computed a cluster statistic by adding up all $t$-values within the cluster.

Cluster statistics were also computed for all 100 label-shuffled classifier traces obtained for each lead, producing a reference distribution of cluster statistics. The cluster statistics from the intact dataset were then compared against this reference distribution using a one-tailed rank test, producing a $p$-value for each cluster. The $p$-values entered a FDR correction procedure with $q = 0.10$[69]. Leads of which at least one cluster survived FDR correction are shown in Fig. 2a; only these leads were included in subsequent analyses.

**Classification within task dimensions.** In order to identify similarities and differences between decision processes for stimulus and task characteristics, we retrained the left/right classifiers and identified temporal clusters for subsets of trials. In Fig. 3, the averaged decoding traces of left/right classification on color rule trials only and orientation rule trials only are shown. We performed cluster-based statistics[68] to test whether these traces differed from each other, for every brain region that had three or more significant leads for both of the rules. First, we performed a two-tailed $t$-test for every time point, identified clusters of significant $t$-values at $\alpha = 0.05$, and determined a cluster $t$-statistic by summing all $t$-statistic values within the cluster. We then randomly shuffled and divided the classifier traces from the color and orientation rules, and performed cluster analysis on this shuffled dataset. We repeated the shuffling procedure 500 times, yielding a reference distribution of cluster $t$-statistics, against that we tested the cluster statistics obtained from the color versus orientation rule comparison using a one-tailed rank test. We performed FDR correction on the resulting $p$-values ($q = 0.10$)[69]. Of the 19 identified clusters, none survived cluster comparison and FDR correction.

Figure 3 also shows the number of leads selected in each brain region by the color rule only and orientation rule only classifiers. We compared these lead counts to lead counts from ten classifiers trained on randomly subsampled datasets with half of the total trial count. This subsampling compensated for a possible reduction in sensitivity due to lower trial count when only including trials from one rule. For every brain region, we compared the difference between color and orientation rule lead counts to the 45 lead count differences from the subsampled data using a one-tailed Wilcoxon rank-sum test with $\alpha = 0.05$. The resulting $p$-values were corrected for multiple comparisons using a FDR correction procedure ($q = 0.10$)[69]. We performed the same analysis for congruent and incongruent trials (subsampled to 30% to match the number of congruent trials), red versus blue trials and horizontal versus vertical trials, the results of which are given in Supplementary Figs. 10 and 11.

**Single-trial classifiers.** For the selected leads, we trained an additional classifier using a leave-one-out cross-validation approach, i.e., the classifier was trained on all-but-one trials, and tested on the left-out trial, and this was repeated for all trials. This provided us, for each time point in the trial, with the distance from the hyperplane separating left from right button presses trained on the same time point from all other trials, a measure known as the decision value ($D$-value). The whole procedure was repeated ten times to minimize the effect of initial conditions of classifier training and performed independently for each time point in the trial[66]. This provided us with a $D$-value time trace for each trial (averaged across the ten

repetitions), thus allowing us to account for any trial-to-trial differences in the processes leading up the response. To avoid circularity in our analyses, we did not analyze the amplitude of $D$-values, as we previously selected the included leads based on the significance of their $k$-fold cross-validation classifier traces (see "Left/right and baseline classifiers" section). Instead, we only report the onset times of $D$-value peaks, and included all trials for which a peak could be detected (see next section).

**Peak onset detection.** For both spectral power contrast and classifier decision values, we aimed to determine the onset of activation/classification for each individual trial. To identify that onset, we used the following procedure:
Step 1: find the highest value in the trace, with the additional requirement for spectral power contrasts of a minimum height of $p_{thres}$ for the trial to be included in further analyses. We set $p_{thres} = SD$, i.e., to 1 standard deviation from the lead mean, computed across all trials. For the spectral power contrast, activation could present as positive or negative deflections from 0. We therefore identified both peaks and troughs in the power contrast traces and analyzed the timing of the peak/trough with the biggest deviation from 0;
Step 2: smooth the trace using a block function five time steps wide (50 ms);
Step 3: working from the time of the peak backward from the peak to the start of the trace, find the first value of the smoothed trace that meets the following requirements: (1) smaller than half of the peak value; (2) derivative smaller than threshold $d_{thres}$, remaining below this threshold for at least $n_t$ time points. In this manuscript, we used $d_{thres} = \frac{1}{peakheight} \tan \frac{5\pi}{180}$, i.e., a derivative falling below five degrees, and $n_t = 3$ time points, or 30 ms.

**Pairwise time difference and bootstrapping.** With the onset of the left/right contrast detected for every trial on every lead, we set out to order the leads according to their onset times. We took two different approaches, both illustrated in Fig. 4a. In the first approach ("average of onsets"), we aggregated all onset times of all leads in one brain area and computed the average onset. The average onsets were then compared for all pairs of brain areas, creating a complete time difference matrix. In the second approach ("trial-by-trial"), we instead compared the onset times for every pair of leads on a trial-by-trial basis, producing a time difference per trial and per lead pair. All single-trial onset time differences were then averaged across all trials and pairs representing a particular pair of brain areas, producing a time difference matrix that only included pairs of areas that were recorded together in one or more of the patients. To ensure statistically sound results, we only included areas with three or more leads in the described analyses.

To test the stability of our time difference estimates, we performed bootstrapping for all temporal difference analyses. For each bootstrap, the data obtained from one recording lead were removed and the time difference matrices were recomputed and used in the analyses described in the following paragraphs. Bootstrapping was repeated until all leads had been removed from the dataset once, producing a total of 86 bootstraps.

**Reconstruction of timeline (MDS) and bootstrapping.** We then converted the time difference matrices, which only contain relative differences in time, back to a timeline, by using a MDS approach. We used the following iterative procedure to reconstruct the timeline:
Step 1: assign all brain areas to a random point in time $T_{init}^i$;
then, for every brain area $i$:
Step 2: compute the error $D_t^j$ (in seconds) between the time difference on the timeline with the desired time difference from the matrix for every other brain area $j$.
Step 3: compute the desired update due to each brain area $j$, as follows:

$$\Delta t^j = \beta \, \mathrm{sgn}\left(D_t^j\right) \left(\exp^{\left|\frac{D_t^j}{\sigma_{ij}}\right|} - 1\right),$$

where $\beta$ is a learning parameter, here set to 0.9, and $\sigma_{ij}$ is the weight (in seconds) assigned to the update. For the trial-by-trial method, we used the standard deviation of difference distribution as weight, while all weights were set to 1 when using the average onset approach.
Step 4: update the location of area $i$ on the timeline: $T_{new}^i = T_{old}^i - \frac{1}{N_a - 1} \sum_j \Delta t^j$,
where $N_a$ is the number of brain areas.
Step 5: repeat Steps 2–4 for all brain areas $i$, until the relative locations of the brain areas, i.e., the ranks, have not changed for five consecutive iterations, or the maximum number of 1000 iterations is reached.

This timeline reconstruction method was used for both the average onset and the trial-by-trial derived time difference matrices, as well as their bootstrapped matrices. The algorithm converged in all cases, typically in <50 iterations. The timeline reconstruction does not provide the time relative to the response, but only relative to other areas, as a uniform shift of all coordinates still yields the same difference distribution. To aid comparison, we translated the reconstructed timelines such that 0 represented the onset of motor cortex (BA4). Note that the bootstraps do not have to be centered around the data from the full dataset, as the MDS approach can shift them relative to the main timeline.

**Rank conjunction and rank clustering score**. In Fig. 5, the sequences of the two pairwise time difference methods (average onset and trial-by-trial) and the two signals (spectral power contrast and classifier decision values) are combined. To this end, we determined the rank order for all 86 bootstraps of each of the four methods and combined these data points into one rank distribution for each brain area.

To assess whether the conjunction between methods was significant, we computed, for every brain area, the rank clustering score $C$ by calculating the average distance between all data points obtained from bootstrapping for all four methods (see "Pairwise time difference and bootstrapping" section), i.e.,: $C = \frac{1}{4N_{boot}}\sum_{k,l}(r_k - r_l)$, with $N_{boot} = 86$ being the number of bootstraps and where $r_k$ and $r_l$ is the rank of datapoint $k$ and $l$, respectively, as obtained with the procedure described in the previous section. We then compared the rank clustering score of each area with rank clustering scores of 500 randomly drawn datasets, containing the same number of data points as the original data. Brain areas of which the average distance $C$ was significantly smaller ($\alpha = 0.05$) than that of the randomized dataset were said to show significant rank clustering across the four methods.

**Software**. All analyses were performed using MATLAB R2018a (The Mathworks, Inc.). Unless otherwise stated, built-in functions were used. Where external functions were used, those functions have been referenced in the relevant sections of the "Methods".

**Reporting summary**. Further information on research design is available in the Nature Research Reporting Summary linked to this article.

## Data availability
The following (fully anonymized) data underlying the results presented here have been made publicly available (https://doi.org/10.6084/m9.figshare.c.4805487; ref. [70]): (1) the across-trial classifier performance for all leads ($t$-values), for all trials combined, as well as for rule, congruency, and stimulus features separately and their label-shuffled reference distributions; (2) the single-trial classifier performance ($D$-values) for the significant leads; and (3) the power spectra for the significant leads. Other data can be provided upon reasonable request, provided doing so does not violate data protection and consent restrictions. A reporting summary for this article is available as a Supplementary Information file.

## Code availability
The custom written functions used to perform peak onset detection, MDS, and rank statistics are available on (https://github.com/marijeterwal/seq-reconstruct]).

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

## Acknowledgements

This work is supported by funding from the European Union's Seventh Framework Programme (FP7/2007-2013) grant nr. 600925 (NeuroSeeker), ERC Advanced grant nr. 323606 (Parietalaction), and European Union's Horizon 2020 Research and Innovation Programme grant nr. 720270 (HBP SGA1) and grant nr. 785907 (HBP SGA2). The authors thank Maria Wimber and Juan Linde Domingo for their helpful suggestions during the early stages of data analysis.

## Author contributions

M.t.W., P.C., G.A.O., and P.H.E.T. designed the study; V.P., G.L.R., and I.S. collected the data; M.t.W., A.P., P.C., and P.A. analyzed the data, and M.t.W., P.A., G.A.O., and P.H.E.T. wrote the manuscript.

## Competing interests

The authors declare no competing interests.
