## [Peer Review File · Nature Communications]

Reviewers' comments:

Reviewer #1 (Remarks to the Author):

Human stereoEEG recordings reveal network dynamics of decision-making in a rule-switching task

This is an exciting study of the neural correlates of decision making in the human brain. The authors record intracranial field potential signals from 6 epilepsy patients while the patients perform simple color or orientation discrimination task with a cue dictating which task is to be performed changing randomly every so often. The results show that two regions within parietal cortex (rostral inferior parietal and parietal opercularis) are particularly activated during the task. The authors propose that the decisions could be implemented in parietal cortex and then submitted to motor cortex for behavioral execution. The results and analyses are interesting and will help the field better understand the dynamics underlying the complex processes of decision making. Mostly, I was excited about the data and wanted to see more details, as described below.

The authors argue in lines 118-120 that accuracy was similar when comparing different rules, colors, orientations, and congruency (I did not understand congruency, see question below). If I understand correctly, they use this similarity to justify lumping everything together when examining the neurophysiological data. I am not sure that I follow the logic here. Let's take one of these comparisons as an example. Let's say that behaviorally, the accuracy is the same for the color rule and the orientation rule. This does not imply that the neural representation of those rules should be the same. For example, one may imagine that color sensitive neurons in area V1 or V4 might respond differentially to one color or the other, or that they may be modulated by attention to color (and orientation tuned neurons in V1 would respond differently to horizontal versus vertical and be modulated differentially by attention to the orientation rule). Thus, the behavioral similarity in accuracy does not seem like a good justification to lump everything together in the analyses of the neural data. It would be interesting to report whether there are differences in the neural responses across rules, colors, orientations, probably differences between rules is the most interesting condition here. This involves a bit more work in terms of analyses but the authors are experts in conducting this type of computational analyses and this quantification should not bring any challenge, if I understand the task and data correctly.

In Figure 2 the authors examine a period between 0.5 s before and 0.15 s after the motor response. Again, why not report other time periods? It seems that the authors are particularly interested in decision making. Based on 1d-e, the reaction times are about 0.5 s. Thus, the analysis window (on average) starts at the offset of the stimulus. Why not look at data during presentation of the stimulus also? Is there any evidence that the decision could not be happening during the 300 ms of stimulus presentation? Related to this, the authors align the neural signals to the motor response. Why not align the responses also to stimulus onset? Is the decision better understood when aligning the responses to the motor response than to the stimulus onset? This is not an obvious point and it would be interesting to show whether this is the case or not (for an example discussion of this point in a different type of experiment see Tang et al eLife 2016)

Also, in Figure 2 the authors and perhaps in subsequent figures (I was not sure), the authors subtract ipsilateral neural responses from contralateral neural responses. Are the neural responses (particularly those in parietal cortex recorded in the current study) lateralized to justify this step? It would be nice to show this lateralization before reporting the subtracted data averaged across all sorts of variables (rules, electrodes, subjects, etc.)

What are chance levels for D values? I assume 0 but this was not stated clearly (what is a D value anyway? Is this the number of trials correctly classified?). What is the variation in D values for randomly shuffled labels? Answers to these questions would help us interpret the results in Fig. 3. Given the noise and variability in the D versus time curves, how reliable are the time differences? If I interpret correctly the individual bootstrapped dots in 3b i and 3b ii, then the distribution of those dots seems to be different from 0 for some of the areas (by about 10 to 40 ms). Is this the correct interpretation of the findings?

Line 108: “combined of all subjects combined”. Too many “combined”.

What exactly does “congruency” mean line 118 and Figure S2? Is this referring to whether the color of the stimulus is the same as the one in the border or not? I could not find a definition of this in the text.

Line 90: “local field potentials”. There is a whole family of “field potential” signals and there is no clear consensus in the field about their nomenclature. I prefer to restrict the use of LFP to signals recorded from high impedance microwires, probably spanning several hundred microns, and to use the term “intracranial field potentials” for the type of data recorded here with low impedance electrodes, probably spanning several millimeters. Of course, the word “local” is arbitrary and could refer to any scale.

Reviewer #2 (Remarks to the Author):

I looked forward to receiving this manuscript and read it with considerable interest. There are many strengths to the manuscript especially the stereoEEG. However, my enthusiasm for the manuscript is limited by the scholarship and I am not sure why such a complex design was used.

It is simply intellectually lazy and misleading to declare in the abstract that the regions of the brain involved in the decision process are not well understood. This strikes me as the level of discourse of an undergraduate paper. There is a wealth of literature on decision making in humans using EEG and fMRI. The authors are dismissive of anything that does not involve intracranial recording because of low spatial or temporal resolution. That’s just ignorance and if we believe the authors, we should ignore the

majority of the human neuroscience literature. All techniques including EEG, fMRI, stereo EEG, LFP, single unit recordings are spatially and temporally filtered representations of the underlying dynamics. No one technique can ever give us a complete picture. For example, despite its slow temporal resolution and indirect measurement of a blood oxygenation signal, only fMRI gives us a complete picture of activation throughout the brain. No other technique can make this claim. Any intracranial technique involves selective placement of electrodes. The results presented in this paper are the luck of the draw in the placement due to surgical requirements and no more than that.

Mike Shadlen and others have published 100's of papers using single unit recordings in monkeys to investigate decision making. The author's level two criticisms at the animal literature. First, the recording area is limited. We can see clearly in Figure 2 that they have only sampled a small portion of the cortex. I would argue the same is true about stereo EEG vis a vis fMRI. There are always trade offs. The second criticism is that the tasks used are too simple, and a more complex task will elucidate the mechanism. I strongly disagree with this claim. For us to understand the results of the stereo EEG studies, which involve a new kind of signal, we need to understand simple tasks first.

In the task they present, there is actually a mixture which makes it difficult to know what the subject is considering. In Fig 1a, we can see that red + horizontal and blue + vertical produce the same response no matter what the cue is. Therefore the cue is only relevant to one of the responses and not the other in either cueing condition. This makes the behavioral data very difficult to model, because there is an odd mixture of a 2AFC task and a 4AFC task. I really don't understand what Panel e is showing as an interaction and the text did not explain it clearly. I am guessing it is saying the subjects are slower at the 2 conditions that have 4AFC properties than 2AFC, but this is not surprising.

I thought the analysis of onset times was very clever. However, for this to be valid, they need to avoid double dipping and segregate part of the data for onset time analysis, and don't use it for electrode selection.

It would help a lot if they did a simple 2AFC task using this technique and use a task that can be easily modeled. For example any 2AFC task can be modeled with a diffusion process which allows for identifying which brain regions are involved in evidence accumulation towards one or more decisions. It would have been much more impactful if the experiment had been conducted in this way. I think they could analyze their data in this manner.

Point-by-point responses to reviewer comments for resubmission of
‘Human stereoEEG recordings reveal network dynamics of decision-making in a rule-switching task’:

Reviewer #1:

1A. *‘The authors argue in lines 118-120 that accuracy was similar when comparing different rules, colors, orientations, and congruency (I did not understand congruency, see question below). If I understand correctly, they use this similarity to justify lumping everything together when examining the neurophysiological data. I am not sure that I follow the logic here. Let’s take one of these comparisons as an example.*

Let’s say that behaviorally, the accuracy is the same for the color rule and the orientation rule. This does not imply that the neural representation of those rules should be the same.

For example, one may imagine that color sensitive neurons in area V1 or V4 might respond differentially to one color or the other, or that they may be modulated by attention to color (and orientation tuned neurons in V1 would respond differently to horizontal versus vertical and be modulated differentially by attention to the orientation rule). Thus, the behavioral similarity in accuracy does not seem like a good justification to lump everything together in the analyses of the neural data. It would be interesting to report whether there are differences in the neural responses across rules, colors, orientations, probably differences between rules is the most interesting condition here.

This involves a bit more work in terms of analyses but the authors are experts in conducting this type of computational analyses and this quantification should not bring any challenge, if I understand the task and data correctly.’

Response: For this manuscript, our primary interest was in identifying and analysing decision processes that are stimulus independent and potentially rule independent. We assumed that by looking at the outcome, i.e. left versus right button presses, across all conditions we would be able to ‘average out’ stimulus or rule dependent processes and leave us with the decision information only. The behavioural similarities, for both accuracy and reaction times, across stimulus characteristics, rules and congruency conditions, supported the assumption that there are no marked differences in output generation (for example, no biases towards one of the two choices) across stimulus characteristics and rules. However, we agree with the reviewer that the behavioural similarities do not sufficiently test this assumption. In the new version of the manuscript we therefore explicitly test the assumption by training the classifier individually for each of the two rules, and on congruent/incongruent, horizontal/vertical and red/blue trials separately. For comparison, we also trained a classifier on a subsampled data sets matching the trial counts in these new classifiers, i.e. with half (or a third of) the number of trials, but containing both rules, congruency and stimulus types. We report that there is no difference in the fraction of channels selected by classifiers trained on color and orientation rules only, neither across regions and within brain regions, nor did we detect any differences between color and orientation rules in the time course of decoding. We only detect small differences in the lead

counts between congruent and incongruent, horizontal and vertical and red and blue trials. We have incorporated these new results in the paper, resulting in a new paragraph (lines 231-249), a new Figure 3 and Supplementary Figures S10 and S11, and a description of the new analyses in the Methods section (lines 610-633). We have added a schematic to summarize these new results together with the results from the timing analysis as a new panel to Figure 5 (Figure 5c).

1B. *'In Figure 2 the authors examine a period between 0.5 s before and 0.15 s after the motor response. Again, why not report other time periods? It seems that the authors are particularly interested in decision making. Based on 1d-e, the reaction times are about 0.5 s. Thus, the analysis window (on average) starts at the offset of the stimulus. Why not look at data during presentation of the stimulus also? Is there any evidence that the decision could not be happening during the 300 ms of stimulus presentation?'*

Response: The reviewer correctly states that the reaction times (RTs) of the patients are around 0.5s. In our task setup, RTs are recorded after stimulus onset, and the cue and stimulus remain on the screen until the response of the patient. This means that the analysis window of 0.5s before response, on average, includes the entire period between stimulus onset to response, and hence includes the period of early visual processing of the stimulus (see also answer to comment 1C). We therefore think that our data indeed show what the reviewer asks and that their question might reflect a misunderstanding caused by the schematic of the task in Figure 1. We therefore made changes to Figure 1a and 1b and clarified the description of the task in the main text (lines 107-108).

1C. *'Related to this, the authors align the neural signals to the motor response. Why not align the responses also to stimulus onset? Is the decision better understood when aligning the responses to the motor response than to the stimulus onset? This is not an obvious point and it would be interesting to show whether this is the case or not (for an example discussion of this point in a different type of experiment see Tang et al eLife 2016)'*

Response: In the manuscript, we decode the response (left vs right button press) to study the decision process, as opposed to evidence for stimulus features. We expect that the representation of the left/right decision is more strongly locked to the response, i.e. less variable relative to the response time, than it is compared to the stimulus onset, because there is no direct mapping from stimulus feature (red/blue, horizontal/vertical) to response due to the rule-switching nature of our task. This expectation is supported by the results from Tang et al., eLife, 2016. Figure 2 by Tang and colleagues demonstrates that response-conflict related signals in a Stroop task, another response-related process, are more strongly locked to the response (Figure 2D-F in their paper) than to stimulus onset (Figure 2A-C). To confirm this expectation in our dataset, we have retrained the left/right classifiers on stimulus-locked trials (0.5 s before stimulus onset to 1 s after stimulus onset). We added the stimulus-locked classifier t-values averaged across leads for the 12 brain regions we previously identified using the response-locked analysis to the Supplementary Information in Figure S5. As expected and in line with the results from Tang et al., the t-values were lower for stimulus-locked than for the response-locked data, suggesting the decision representation we decode is indeed more locked to the response. Though the stimulus-locked data did show successful decoding for the identified regions, we argue that the response-locked analysis is the most appropriate approach for our research question.

1D. *'Also, in Figure 2 the authors and perhaps in subsequent figures (I was not sure), the authors subtract ipsilateral neural responses from contralateral neural responses. Are the neural responses (particularly those in parietal cortex recorded in the current study) lateralized to justify this step? It would be nice to show this lateralization before reporting the subtracted data averaged across all sorts of variables (rules, electrodes, subjects, etc.)'*

Response: Figure 2b shows the contra-ipsilateral contrast averaged across subjects. As each subject only had electrodes implanted in one hemisphere, the contra-ipsilateral contrast made it possible to pool data across subjects with electrodes in opposite hemispheres. Note that within subject, the contra/ipsi contrast is identical to the left/right contrast used for the classifier. We used this equivalent left/right contrast for the power spectra of each individual subject to identify regions that are involved in the decision process and assess their timing. As opposed to a general task-related activity that is identified by a baseline-contrast, the left/right contrast specifically identifies differences in activation for the two answer options, which is what would be expected for the representation of a decision. This contrast captures both potential lateralization of activation, as well as potential differences in activation (e.g. amplitude differences between correct and incorrect answer options). It is important to note that because none of the patients had bilateral implantations we cannot assess lateralization directly (i.e. within a trial). Instead, we contrasted the power spectra against the pre-cue baseline period and added the average spectra to the supplemental information for contralateral and ipsilateral trials separately (Supplementary Figure S7). We also re-computed the correlations for the ipsilateral and contralateral trials separately (Supplementary Figure S9). We added a clarification to the main results text in lines 206-211 and added the additional baseline analysis to the Methods section (lines 564-566).

1E. *'What are chance levels for D values? I assume 0 but this was not stated clearly (what is a D value anyway? Is this the number of trials correctly classified?). What is the variation in D values for randomly shuffled labels? Answers to these questions would help us interpret the results in Fig. 3.'*

Response: The D-value, or decision value, is the distance of the left-out trial to the hyperplane separating left from right as trained on all other trials. The minimum possible distance is indeed 0. We have added an explanation about D-values in the main text (lines 270-273) and the Methods section (lines 636-638).

It is however important to note that we do not analyse the amplitude of the D-value, only the timing of it. We do not assume that the D-values are significant on individual trials (this is a conservative approach aimed to prevent bias in the timing analyses) and do not analyse the D-value (see also the response to comment 2E); this would be double dipping, because we have selected the channels based on their across-trials performance (Figure 2). We have added a sentence explaining this approach to the Methods section (lines 642-646).

1F. *'Given the noise and variability in the D versus time curves, how reliable are the time differences? If I interpret correctly the individual bootstrapped dots in 3b i and 3b ii, then the distribution of those dots seems to be different from 0 for some of the areas (by about 10 to 40 ms). Is this the correct interpretation of the findings?'*

Response: The dots in Figures 4 and 5 (previously Figures 3 and 4) indeed represent the uncertainty in the onset timing of the regions, i.e. a wide dot cloud around the line represents a high uncertainty and a narrow dot cloud a low uncertainty. The multidimensional scaling procedure (used to place the regions on a timeline) can shift the dots away from 0, especially when uncertainty is higher. We have added a clarification to the Methods section (lines 706-708). The dot clouds combine uncertainty from both the D-value (onset detection) as well as uncertainty from the placement of the regions onto the time axis. For the uncertainty in the D-value onsets alone we refer to Supplementary Figure S12, which shows the histograms of onset times per lead. The bootstrapping that was done to create the dots in Figures 4 and 5 was performed on the level of leads, not trials, and will therefore be a more conservative estimate of certainty.

1G. *'Line 108: "combined of all subjects combined". Too many "combined".'*

Response: We have changed this to: *'left and right button presses of all subjects combined'*.

1H. *'What exactly does "congruency" mean line 118 and Figure S2? Is this referring to whether the color of the stimulus is the same as the one in the border or not? I could not find a definition of this in the text.'*

Response: We have added an explicit definition of congruency to the main text in lines 103-104 and the Methods section in lines 499-501.

1I. *'Line 90: "local field potentials". There is a whole family of "field potential" signals and there is no clear consensus in the field about their nomenclature. I prefer to restrict the use of LFP to signals recorded from high impedance microwires, probably spanning several hundred microns, and to use the term "intracranial field potentials" for the type of data recorded here with low impedance electrodes, probably spanning several millimeters. Of course, the word "local" is arbitrary and could refer to any scale.'*

Response: We use the term 'local field potential' here as described in the paper by Buzsaki, Anastassiou & Koch, Nature Reviews Neuroscience, 2012. We have added a reference to this paper in line 96.

Reviewer #2:

2A. *'It is simply intellectually lazy and misleading to declare in the abstract that the regions of the brain involved in the decision process are not well understood. This strikes me as the level of discourse of an undergraduate paper. There is a wealth of literature on decision making in humans using EEG and fMRI. The authors are dismissive of anything that does not involve intracranial recording because of low spatial or temporal resolution. That's just ignorance and if we believe the authors, we should ignore the majority of the human neuroscience literature. All techniques including EEG, fMRI, stereo EEG, LFP, single unit recordings are spatially and temporally filtered representations of the underlying dynamics. No one technique can ever give us a complete picture. For example, despite its slow temporal resolution and indirect measurement of a blood oxygenation signal, only fMRI gives us a complete picture of activation throughout the brain. No other technique can make this claim. Any intracranial technique involves selective placement of electrodes. The results presented in this paper are the luck of the draw in the placement due to surgical requirements and no more than that.'*

Response: In the manuscript we present two previously unreported decision-related brain areas. Our results raise the question why these regions have not been identified before and we elaborate on this question in the Discussion section. We agree with the reviewer that the second sentence of the abstract and two sentences in the introduction (lines 69-73) took a short-cut in this explanation. We have removed these sentences and changed the introduction to be more in line with the tone of the remainder of the paper. We trust that the analysis we provide in the Discussion section of the paper (line 437-449) of possible reasons for why fMRI and EEG/MEG might have missed the reported regions gives a balanced representation of the field.

2B. *'Mike Shadlen and others have published 100's of papers using single unit recordings in monkeys to investigate decision making. The author's level two criticisms at the animal literature. First, the recording area is limited. We can see clearly in Figure 2 that they have only sampled a small portion of the cortex. I would argue the same is true about stereo EEG vis a vis fMRI. There are always trade offs.'*

Response: We fully agree with the reviewer on these points and have removed the sentences in question (see answer to comment 2A). We state the limitation of coverage in our study explicitly in lines 406-409 of the discussion.

2C. *'The second criticism is that the tasks used are too simple, and a more complex task will elucidate the mechanism. I strongly disagree with this claim. For us to understand the results of the stereo EEG studies, which involve a new kind of signal, we need to understand simple tasks first.'*

In the task they present, there is actually a mixture which makes it difficult to know what the subject is considering. In Fig 1a, we can see that red + horizontal and blue + vertical produce the same response no matter what the cue is. Therefore the cue is only relevant to one of the responses and not the other in either cueing condition. This makes the behavioral data very difficult to model, because there is an odd mixture of a 2AFC task and a 4AFC task.'

Response: It is important to note that the rule-switching task used here is in fact a 2AFC task, as on every trial the subject is presented with only one stimulus, and the only possible answers for each stimulus are left or right. Instead, what changes between trials is the mapping of the stimulus to the required output. The reviewer correctly noted that for two stimuli, namely red-horizontal and blue-vertical, this mapping was always the same (congruent trials), while for red-vertical and blue-horizontal, the mapping depended on the rule (incongruent trials). However, the possible answers did not change, and therefore the task maintained a 2AFC setup on all trials. We think that the confusion regarding the task setup stems in part from the schematics in Figure 1a and 1b, see also comment 1B, and we have therefore improved this figure. We have also clarified the definition of congruent and incongruent trials in the main text (lines 103-104) and in the Methods section (lines 499-501).

Furthermore, in the new version of the manuscript we explicitly compare the decision processes for the two rules and for congruent and incongruent trials separately (as well as for stimulus features), see also the response to comment 1A. We report that we did not detect any differences in the brain areas involved in decision making for different rules and no differences in decoding time traces for the two rules. These results suggest that we have identified a rule-independent decision-related process that is independent of evidence accumulation for specific stimulus features (i.e. red versus blue) and rule-induced top-down modulation. We added a schematic summarizing these results to Figure 5c.

2D. *'I really don't understand what Panel e is showing as an interaction and the text did not explain it clearly. I am guessing it is saying the subjects are slower at the 2 conditions that have 4AFC properties than 2AFC, but this is not surprising.'*

Response: The goal of the old version of Figure 1e was twofold: 1) it demonstrated the difference in reaction times on color and orientation rules (main effect) and 2) it demonstrated an interaction between stimulus color (red or blue) and rule, i.e. that red stimuli were answered quicker for the color rule, while blue stimuli were answered quicker for the orientation rule (even though this feature was irrelevant in determining the correct response!). We think the latter represents a priming effect, as the color rule was cued by a red outline, while the orientation rule was cued by a blue outline (see lines 493-494). This result speaks against a dominant and exclusive influence of rule or congruency alone, but this interaction is not the focus of the paper. Now that we have added explicit tests of rule, congruency and stimulus features on the decision process (Figure 3, S10, S11), we have decided to simplify panel 1e and only show the main effect of rule on RT. We have kept the analyses of interactions between rule and stimulus features in RTs in the Supplementary Figure S2.

2E. *'I thought the analysis of onset times was very clever. However, for this to be valid, they need to avoid double dipping and segregate part of the data for onset time analysis, and dont use it for electrode selection.'*

Response: Like the reviewer we appreciate that care has to be taken when selecting part of the dataset for further analyses using the same measure. We have therefore been very careful not to analyse the amplitudes of the classifier performance in the selected leads, as the amplitude is the measure we used for selection of the leads in the first place. Instead, we

only analyse the timing of decoding. The identification of onset of successful classification is an additional analysis that is independent of the amplitude of classification, but cannot be performed on leads without successful classification. We therefore feel that we have adequately taken care in our analyses to avoid double dipping. We have added a sentence explicitly explaining our approach to the Methods section in lines 642-646.

2F. *'It would help a lot if they did a simple 2AFC task using this technique and use a task that can be easily modeled. For example any 2AFC task can be modeled with a diffusion process which allows for identifying which brain regions are involved in evidence accumulation towards one or more decisions. It would have been much more impactful if the experiment had been conducted in this way. I think they could analyze their data in this manner.'*

Response: We like to note that in fact, on any given trial in the rule-switching task, only one rule is in effect and only two answer options (left or right) are available, and as such, the task used here is in fact a 2AFC task (see also answer 2C). We use an explicit decoding of the left/right choice to identify regions involved in the decision. This analysis approach can be performed even when evidence accumulation for sensory features may not have been recorded fully, as is the case in our dataset (we have no to limited coverage of early visual areas and IPS). Though used here for the response, not the stimulus, the decoding approach has a strong parallel with the evidence accumulation model, as it captures the difference in support for the two response options over time. We have now added detailed analyses showing that the decision process we identified is common to both rules (Figure 3). The only difference between the rules is the type of visual information that is informative for the decision. This suggests that although the sensory evidence is different, the accumulation of evidence for the answer options is the same between rules (Figure 5c).

We hope that the additional results (Figure 3) and the new summary of our results in Figure 5c further clarify these points. We would like to refer the reviewer to the answers given in response to comments 1A, 1B and 2C for a more detailed explanation of how we addressed the points raised here.

Reviewers' comments:

Reviewer #1 (Remarks to the Author):

The authors have done multiple analyses and rewritten parts of the text to clarify the text. I think that the current version reads much better and adequately addresses the previous concerns.

I am not quite in agreement with the LFP nomenclature in Buzsaki, Costas, Koch review (local is an ambiguous term that spans orders of magnitude in different scales, yet that is in an excellent review article otherwise). But this is a subtle issue of nomenclature and should not distract from the interesting findings in the current work.

I was intrigued by the new analyses where the authors find no difference in the classifier across different conditions (color, etc). This is quite interesting.

Point-by-point responses to reviewer comments for resubmission of
‘Human stereoEEG recordings reveal network dynamics of decision-making in a rule-switching task’:

2D. *“The interactions between congruency, rules and stimulus were not clear as Reviewer 2 pointed out (also pointed out by Reviewer 1). The authors now simplified the description and removed some of the complexities in the interpretation. This may be seen as sweeping under the rug some of the difficult interactions in the task. At the very least, it would be useful to explain what they are simplifying, why, and clearly point out the more detailed discussion of these points in the main text.”*

Response: We feel that it is important to note that the interaction results were not removed from the paper. The interaction is described in the results section of the paper and results from all tests performed on the behavioural data are available in the Supplemental Information. We simplified Figure 1e by only showing the main effect, because the figure with interaction led to unnecessary confusion and served no purpose in the interpretation of the response timings, especially after the introduction of explicit comparisons of the networks involved in each of the rules separately (Figure 3). We have, to clarify the interpretation of the interaction, added 2 sentences to the main text. Lines 130-133 now read: “Reaction times showed an interaction between rule and color of the stimulus (Figure S2; $p = 0.0005$; 2-way ANOVA, $F(1,3124) = 12.12$), probably caused by priming due to the colored rule cue. This interaction is expected to have little effect on the decision-related and response-locked analyses presented in the remainder of the paper.”

2E. *“Most likely the analyses do not involve double dipping as the authors discuss. The only remaining concern could be if there is a correlation between amplitudes and onset timing. The authors state in the response that these are independent metrics but provide no evidence for this alleged independence. In my experience, intensities (amplitudes, firing rates, power) and onset latencies are **not** independent. But it is hard to rigorously assess this from the results presented in the paper without an explicit quantitative assessment of these potential interactions.”*

Response: In general, there are many cognitive and physiological reasons why amplitudes and onset latencies can be negatively correlated when comparing responses across trials/stimuli, especially in sensory regions, e.g. earlier and stronger responses for clearer stimuli and later and weaker responses for ambiguous stimuli. It is less clear whether such a relationship also holds for responses of different brain regions within one trial (i.e. in response to the same stimulus). There can also be technical reasons for a negative correlation across trials between amplitude and onset latency, for example, a reduction in jitter in response onset for less ambiguous stimuli can increase the signal-to-noise ratio after averaging across trials, leading to a higher amplitude and an earlier onset latency of the averaged response.

In the paper, we select channels based on amplitude obtained using an across-trials classifier result, while we assess the latencies of single-trial classifier results. We now tested the relationship between amplitudes and latencies of both classifiers explicitly in our dataset. We found a no significant relationship when we determined both amplitude and onset latency using the across-trial classifier performance (Pearson correlation; $\rho = -0.0633$; $p = 0.103$; $N = 663$).

The across-trial classifier was used to select significant leads ($N = 95$) for which onset latency was determined using single-trial classifier results. Considering the single-trial onset latency and amplitude (the latency data presented in the paper), yielded a positive correlation between average single-trial onset latency and average single-trial amplitude across leads (Pearson correlation; $\rho = 0.521$; $p < 0.0001$; $N = 95$ leads). Hence, when ordered based on the single-trial latencies (as in Figure 5 of the main paper), the data show an amplitude increase with latency for both the single-trial classifiers and the across-trial classifiers. These results suggest a build-up of evidence across the brain regions involved in decision making. Such a relationship would be worth exploring further in the future, but a detailed analysis of the amplitude relationships is beyond the scope of the current paper.

We have added the amplitude findings to the Supplementary Information in the form of the new figure S14. We have referenced this finding in the results section; lines 341-344 now read: "The late responding BA3a/S1 areas also stand out due to their high decoding performance (Supplementary Figure S4). This suggests that for decision regions, there is little or no evidence that earlier onset is coupled to the highest decoding performance (see also Supplementary Figure S14).

***REVIEWERS' COMMENTS:

Reviewer #1 (Remarks to the Author):

First, I apologize for the delay.

The authors have done a very thorough job to address all the previous questions.

I am still surprised that the authors claim not to find differences based on color, orientation. They do report (small) differences in the lead counts that can be attributed to congruent/incongruent, color and orientation. However, the authors carefully considered this and present new analyses that clearly address this question.

This is an exciting manuscript and it is ready to share with the scientific community at large.